# Multiple Linear Regression Analysis of Canada's Freight Transportation Framework

**Jamileh Yousefi [1,*], Sahand Ashtab [1,*], Amirali Yasaei [2], Allu George [1], Ali Mukarram [1] and Satinderpal Singh Sandhu [1]**

1   Shannon School of Business, Cape Breton University, Sydney, NS B1P 6L2, Canada
2   Faculty of Engineering, Waterloo University, Waterloo, ON N2L 3G1, Canada
*   Correspondence: jamileh_yousefi@cbu.ca (J.Y.); sahand_ashtab@cbu.ca (S.A.); Tel.: +1-(902)563-1227 (J.Y.)

**Abstract:** *Background*: Finding trends in freight transportation activities enables businesses and policy makers to build an understanding of freight transportation patterns and their impact on logistics planning when making investments in a region's transportation infrastructure and intermodal freight transport system. To the best of our knowledge, there is limited literature and data-driven analysis about trends in transportation mode choices and the influencing factors in Atlantic Canada. *Methods*: In this study, a data-driven method has been used to analyze the Canadian Freight dataset to identify trends in transportation activities within Maritime, Canada. Freight transportation mode, product categories, distance, number/weight of shipments, and revenue were examined. *Results*: The results revealed that the top five product categories exported from Atlantic provinces to the rest of Canada, the US, and Mexico are miscellaneous items, food products, forest products, minerals, and other manufactured goods, where Truck for Hire is the most deployed mode of transportation. A multiple linear regression analysis indicated that the weight, distance, and number of shipments are positively and rather strongly correlated with revenue generation. *Conclusions*: This study provides a unique overview of Canadian Freight Analysis Framework (CFAF) data with a focus on maritime activities.

**Keywords:** transportation; Canada; truck; multiple regression; analytics; linear regression

## 1. Introduction

The economic growth of a region is directly linked to its transportation infrastructure, trade, and freight transportation volume. Freight transportation is one of the drivers of supply chain management [1], which involves strategic and tactical decisions concerning the planning and selection of a transportation mode for the transportation of goods and services internationally or interprovincially. The decisions in this regard affect the performance, quality, and cost of inbound and outbound transportation activities. Additionally, it is a crucial factor for prosperity, quality of life, and future development in the region. Hence, exploring patterns is important in the context of investments in regional transportation infrastructure, as pointed out in the Atlantic Canada Transportation Strategy report [2].

### 1.1. Background

In the report by the Atlantic Canada Transportation [2], cost, value, and distance are the three independent factors that have been considered as the main mode choice factors. Air freight has been identified as the preferred mode for time-sensitive goods and low-volume goods, such as medical equipment or precious gems, because their cost per unit is higher. These costs are then absorbed within the sale value of the cargo. In contrast, for unrefined crude oil or iron that have lower per unit weight and are less time-sensitive but higher in volume, slower but higher-capacity transport modes are used.

Hassanzadeh et al. used a SWOT matrix (analysis) to analyze the transportation mode choices in Cape Breton Island, Nova Scotia [3]. An Analytic Hierarchy Process (AHP)

method was used to rank the factors. The analysis suggested that the four important elements of SWOT in the Nova Scotia transportation planning are "roads and highways", "Port of Sydney development plan", "Lack of governance on port of Sydney and lack of co-operation between municipality and senior governments", and "Poor economic conditions impacting transportation development".

Brooks and Trifts [4] examined the mode choice factors for decisions between short sea and trucking options in Atlantic Canada. Three factors of influence, namely, price, transit time, and frequency, were examined. The results showed that shippers' perceptions of the short sea are favorable in the Atlantic market. The study also identified the challenges that need to be addressed for freight by short sea shipping operations to compete effectively with all-truck routes. Tagawa et al. [5] analyzed factors influencing mode-choice decisions in maritime shipment network also pointing out the importance of incorporating qualitative attributes to the analysis.

In another study by Brooks and Frost [6], the opportunities and issues related to the short sea transportation mode in the Atlantic region of North America have been discussed. The study examined the market demand for short sea transportation in four clusters along eastern North America. The results of this study suggested that the distance from Atlantic Canada to Maine is too short to make short sea competitive against trucks. Most shippers in Atlantic Canada have a tight delivery window; therefore, they rely on trucks. Additionally, customs clearance for the short sea mode, which is more difficult for shipping via sea mode than trucks, was identified as another barrier. This study suggested that trucking companies and potential short sea operators should provide an integrated transport package.

### 1.2. Research Contribution

While rail is one of the most important modes of freight transportation in the world, it is unrealistic to assume that all freight could be moved by rail. The mode choice depends on the availability of the transportation infrastructure and the type of goods. Some goods are suitable for air transportation, whereas others are suitable for transportation via rail or truck. However, some goods might be well suited to change from one freight mode to another. The potential for changing the transportation mode depends on the size of the shipment and the distance.

To the best of our knowledge, there is limited literature and data-driven analysis about trends in transportation mode choices and the influencing factors in Atlantic Canada. This paper focuses on the analysis of the Canadian Freight Analysis Framework (CFAF) dataset to identify trends in transportation modes and the movement of certain product types within the Atlantic provinces of Canada. The study investigates the relationships between factors such as transportation modes, product types, revenue generated, and shipment values. The frequency of deployment of different transportation modes (i.e., air, truck, and rail) from and to the Atlantic provinces, as well as the interprovincial shipment of different product types, are also discussed. It is worth mentioning that the availability, consistency, and completeness of data are the major challenges in conducting such a study. The other limitation of this study is the lack of data on the deployment of air and sea transportation modes in some areas.

The structure of this paper is as follows. Section 2 provides an overview of the dataset and methods. In Section 3, the results and discussion are presented. Section 4 discusses the results obtained from the multiple regression analysis. Finally, Section 4 provides the conclusions and future research.

## 2. Materials and Methods

### 2.1. Dataset

The dataset used in the study is the Canadian Freight Analysis Framework (CFAF), which includes 50,210 instances of shipment information between Canada, the US, and Mexico. The data have been collected by Statistics Canada from several sources to create a comprehensive picture of freight flows across North America. The CFAF dataset contains

14 features (year, product type, sub-provincial area (shipment origin), country (shipment origin), sub-provincial area (destination origin), country (destination origin), the aggregate number of shipments, the aggregate weight of shipments (Kg), aggregate revenue earned by carriers, distance (Km), TonneKm, and value) representing three modes (classes) of transportation (air, truck, and rail). Air, trucks, and rails are the modes of transportation used to transport different types of products, including agricultural products, food, minerals, transportation equipment, plastic, and chemical products. Statistics Canada notes that "for air and truck, a shipment represents the movement of a single commodity from an origin to a destination. For rail this represents the number of cars". This can be interpreted as the aggregate number of shipments. which is not equal to the actual total number of shipments; rather, it represents the total number of transported commodities.

There are 596 records of information available regarding the transportation of products from different provinces in Canada to Nova Scotia. The summary information for the total inbound shipments, total TonneKm, total shipment value, and revenue generated by carriers for the province of Nova Scotia are presented in Table 1. The exact number of freight shipments via air is not provided to meet the confidentiality requirements of the Statistics Act. The summary information for the total inbound shipments, total TonneKm, total shipment value, and revenue generated by carriers in the province of New Brunswick between 2011 and 2016 is presented in Table 2.

**Table 1.** Inbound shipments to Nova Scotia from other Canadian provinces.

| Year | Total Number of Shipments | Total TonneKm | Total Shipment Value | Total Revenue Earned by Carriers |
|------|---------------------------|---------------|----------------------|----------------------------------|
| 2011 | 1,253,482 | 7,047,863,934 | CAD 28,031,035,422 | CAD 666,859,119 |
| 2012 | 1,245,306 | 6,689,204,634 | CAD 31,780,643,580 | CAD 687,348,190 |
| 2013 | 1,272,170 | 7,142,635,806 | CAD 30,324,182,403 | CAD 696,063,403 |
| 2014 | 1,273,006 | 6,667,215,334 | CAD 27,955,593,227 | CAD 675,826,517 |
| 2015 | 1,223,374 | 6,988,450,358 | CAD 28,427,268,761 | CAD 631,229,644 |
| 2016 | 1,295,798 | 7,202,189,787 | CAD 32,498,050,364 | CAD 718,559,814 |

**Table 2.** Inbound shipments to New Brunswick from other Canadian provinces.

| Year | Total Number of Shipments | Total TonneKm | Total Shipment Value | Total Revenue Earned by Carriers |
|------|---------------------------|---------------|----------------------|----------------------------------|
| 2011 | 1,234,620 | 7,515,101,458 | CAD 36,430,490,469 | CAD 640,167,051 |
| 2012 | 1,298,787 | 10,631,655,193 | CAD 35,109,593,370 | CAD 826,373,068 |
| 2013 | 1,228,205 | 11,004,461,194 | CAD 32,382,113,308 | CAD 832,571,179 |
| 2014 | 1,306,707 | 12,197,146,309 | CAD 32,156,352,025 | CAD 942,739,350 |
| 2015 | 1,298,803 | 13,227,500,230 | CAD 37,642,259,953 | CAD 963,812,314 |
| 2016 | 1,315,961 | 13,711,836,472 | CAD 37,602,132,494 | CAD 940,842,139 |

The summary information for the total inbound shipments, total TonneKm, total shipment value, and revenue generated by carriers in the province of Newfoundland and Labrador between 2011 and 2016 is presented in Table 3. The summary information for the total inbound shipments, total TonneKm, total shipment value, and revenue generated by carriers in the province of Prince Edward Island between 2011 and 2016 is presented in Table 4.

**Table 3.** Inbound shipments to Newfoundland and Labrador from other Canadian provinces.

| Year | Total Number of Shipments | Total TonneKm | Total Shipment Value | Total Revenue Earned by Carriers |
|------|---------------------------|---------------|----------------------|----------------------------------|
| 2011 | 591,526 | 3,489,099,196 | CAD 17,735,759,898 | CAD 451,043,471 |
| 2012 | 701,395 | 3,611,128,539 | CAD 16,761,951,282 | CAD 495,442,219 |
| 2013 | 880,072 | 2,623,420,883 | CAD 9,273,539,031 | CAD 421,056,463 |
| 2014 | 649,701 | 3,720,469,944 | CAD 12,316,758,786 | CAD 535,362,614 |
| 2015 | 597,303 | 3,067,894,755 | CAD 9,521,704,507 | CAD 467,525,016 |
| 2016 | 577,610 | 3,592,056,138 | CAD 11,776,422,595 | CAD 405,310,459 |

**Table 4.** Inbound shipments to Prince Edward Island from other Canadian provinces.

| Year | Total Number of Shipments | Total TonneKm | Total Shipment Value | Total Revenue Earned by Carriers |
|------|---------------------------|---------------|----------------------|----------------------------------|
| 2011 | 148,638 | 405,460,885 | CAD 2,535,132,534 | CAD 54,494,308 |
| 2012 | 141,950 | 441,455,616 | CAD 3,347,252,745 | CAD 61,442,118 |
| 2013 | 146,927 | 261,411,418 | CAD 2,845,936,534 | CAD 44,070,637 |
| 2014 | 168,967 | 341,713,525 | CAD 3,151,889,070 | CAD 60,807,620 |
| 2015 | 196,585 | 389,635,337 | CAD 4,337,280,456 | CAD 59,685,575 |
| 2016 | 169,878 | 535,876,762 | CAD 3,972,637,034 | CAD 68,011,597 |

*2.2. Methods*

This research included a descriptive analysis of the latest available CFAF datasets. Specifically, this study explored the freight transportation data for Atlantic Canadian provinces with a focus on identifying the trends in transportation activities in terms of transportation modes, product types, revenue generated, and shipment values. Additionally, a multiple linear regression model was used to determine the relationship between freight revenues and other features such as mode, weight, and commodity types.

## 3. Discussion

### 3.1. Analysis of Freight Transportation Mode Choice Factors

In this section, we analyze the transportation mode for the Atlantic region, followed by an analysis of each Atlantic province. The aim of this analysis was to identify the factors that affect mode choice decisions. In addition to identifying these factors, we aimed to discover the potential for shifting freight work from one mode to another. Figure 1 shows the percentage of shipments dispatched from Atlantic provinces based on the transportation mode. As Figure 1 indicates, Truck for Hire is the most widely used transportation mode by the Atlantic provinces over the years. Truck for Hire accounts for 80% of shipments, 19% of goods are shipped by rail, and 1% is shipped by air.

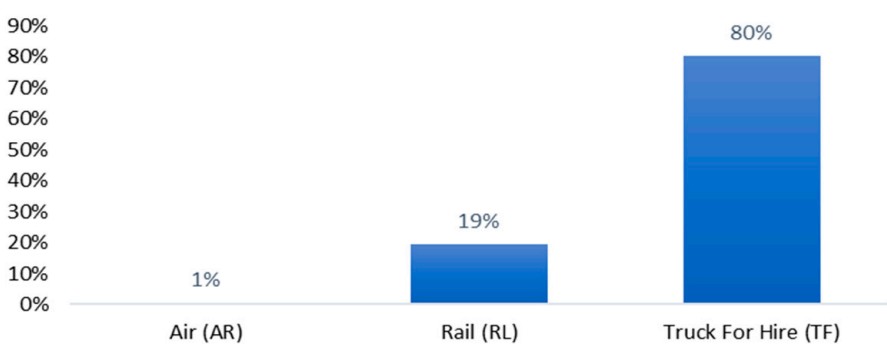

**Figure 1.** Shipments dispatched from Atlantic provinces based on the mode choice.

Figure 2 depicts the comparison of truck services between the Atlantic and the rest of Canada. As shown in the figure, even though the rate of truck services in Canada significantly decreased since 2015, it significantly increased after 2013 in the Atlantic. The increase in using truck service in the Atlantic could be attributed to the closing of the Cape Breton railroad after 135 years [7].

Figure 3 depicts the comparison of air services between Canada's Atlantic provinces and the rest of Canada. As it appears in the figure, in the rest of Canada, there was a significant decrease before 2013, followed by skyrocketing growth after 2013. However, the overall trend looks robust with a slight decrease in air mode choice in the Atlantic over the five years. Similarly to the truck services, the rest of Canada provides more air services than the Atlantic provinces.

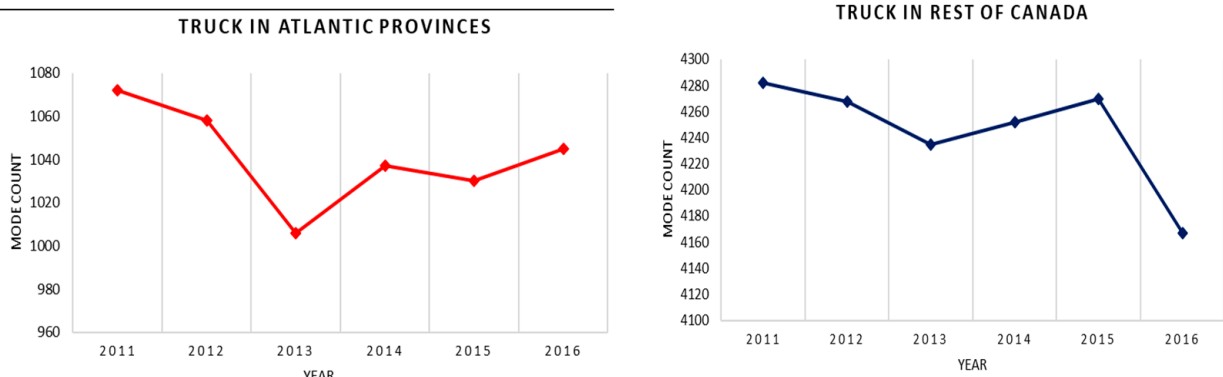

**Figure 2.** Truck service in Atlantic provinces and the rest of Canada.

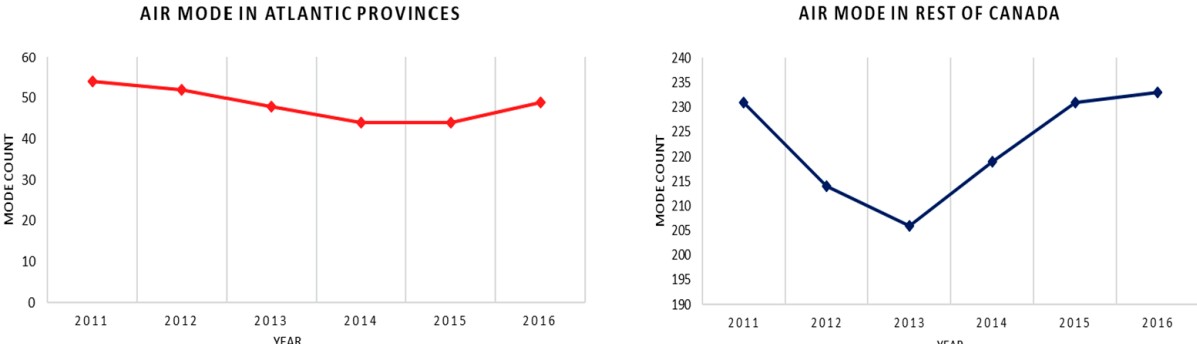

**Figure 3.** Air service in Atlantic provinces and the rest of Canada.

Figure 4 depicts the comparison of rail services between Canada's Atlantic provinces and the rest of Canada. As it appears in the figure, after 2013, all regions of Canada, including the Atlantic, show a continuously decreasing trend in using rail services. The constant trend in the choice of air transportation mode for the Atlantic after 2014 was due to the closing of rail services in Eastern Nova Scotia.

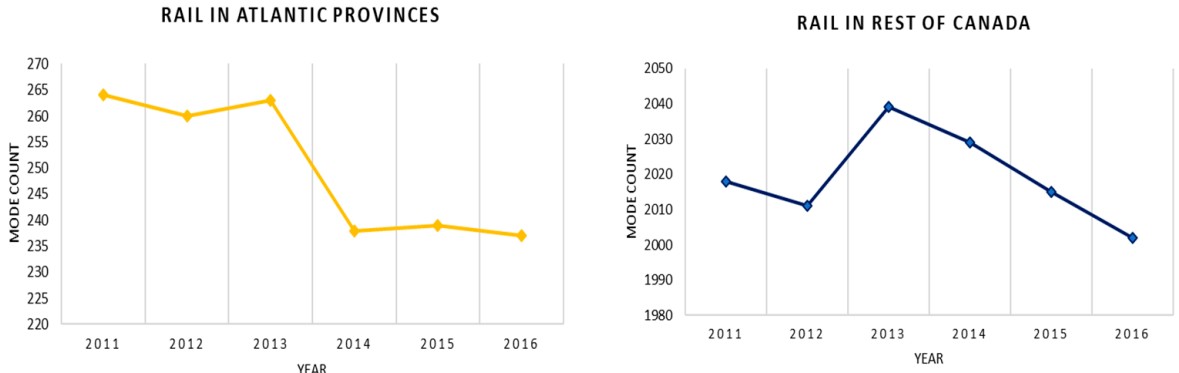

**Figure 4.** Rail service in Atlantic provinces and the rest of Canada.

Figure 5 demonstrates the growth trend of truck services in the Atlantic region. As shown in the figure, the overall trend for Truck for Hire in exports seems to have fluctuated from 2011 to 2016. However, from 2015 onward, this trend seems to be positive. On the other hand, the overall trend of using Truck for Hire mode for importing goods shows a somewhat steady trend, except after 2015.

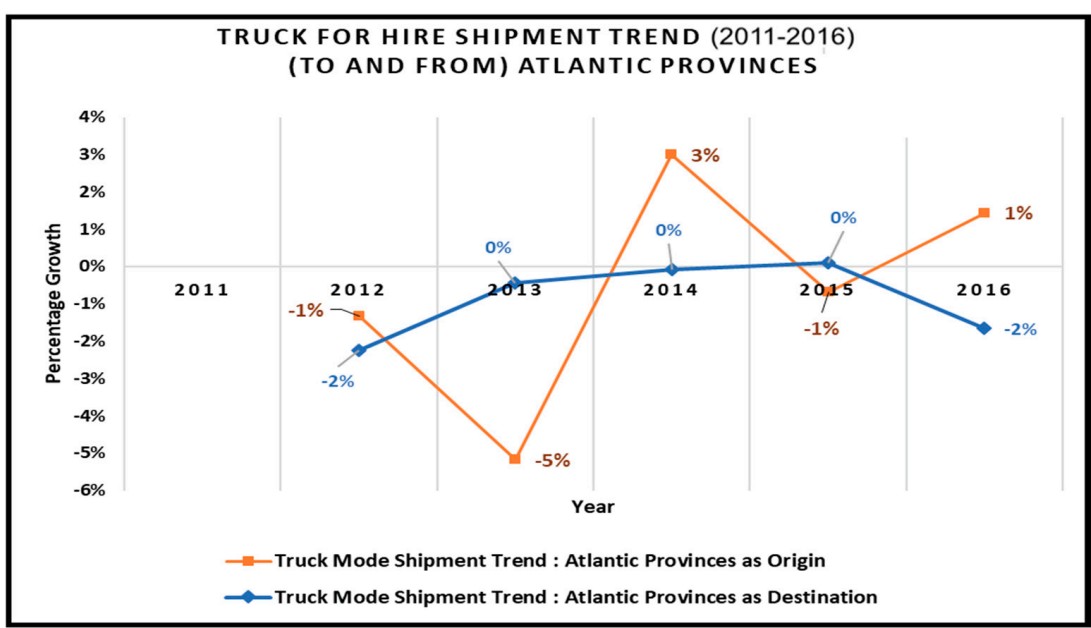

**Figure 5.** Truck for Hire shipment growth in Atlantic provinces.

*3.2. Comparing Atlantic Provinces*

3.2.1. Nova Scotia

The total inbound shipments made by each transportation mode to the province of Nova Scotia are plotted in Figure 6. As illustrated in the figure, the province of Nova Scotia heavily depends on the truck transportation mode for the delivery of inbound shipments. In fact, 92.9% of the total inbound shipments to the province of Nova Scotia are made by using trucks. The breakdown of the total inbound shipments to the province of Nova Scotia is provided in Figures 7–9, which demonstrate the trends in the use of each transportation mode.

While the use of air transportation modes has been decreasing over the years, the figures show that truck and rail transportation modes have supplemented each other. For example, an increase in the use of rail transportation modes led to a decrease in the use of truck transportation modes from 2014 to 2015. It is important to note that the number of shipments reported for the rail transportation mode represents the number of cars and not single commodities, which is the case with the truck transportation mode.

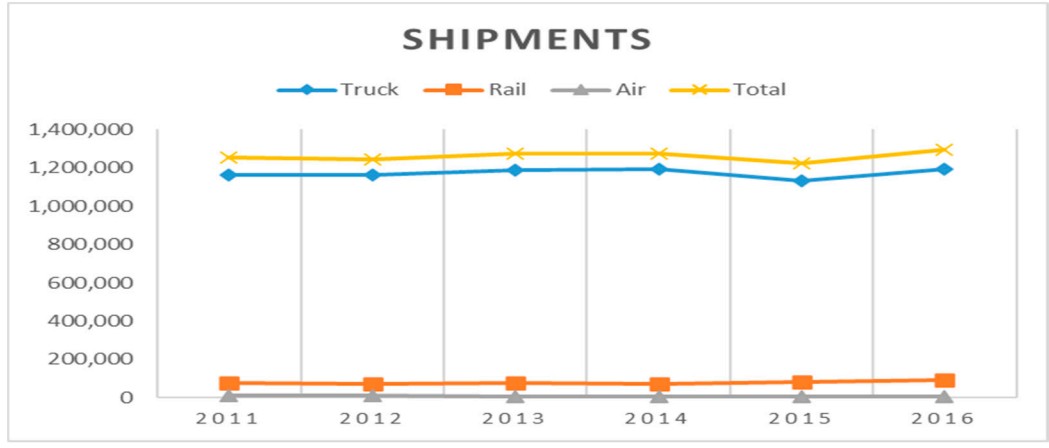

**Figure 6.** Total inbound shipments to the province of NS by different transportation modes.

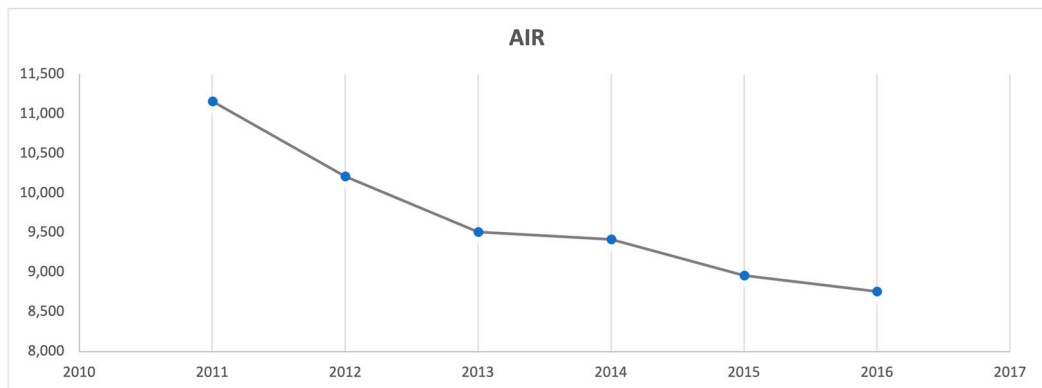

**Figure 7.** Inbound shipments to the province of Nova Scotia via air.

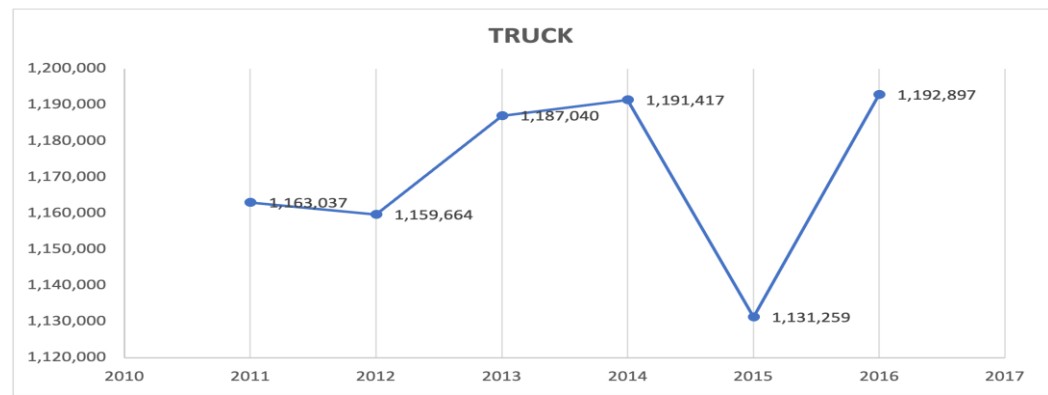

**Figure 8.** Inbound shipments to the province of NS via Truck for Hire.

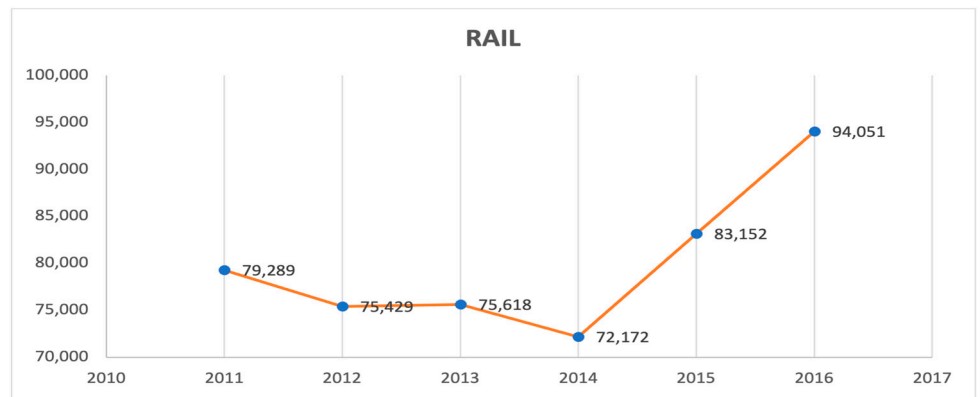

**Figure 9.** Inbound shipments to the province of Nova Scotia via rail.

### 3.2.2. New Brunswick

The total inbound shipments made by each transportation mode to the province of New Brunswick are plotted in Figure 10.

Similar to the province of Nova Scotia, a high volume of shipments to the province of New Brunswick takes place via truck transportation mode. That is, 94.5% of the total inbound shipments to the province of New Brunswick are made using trucks. The breakdown of the total inbound shipments to the province of New Brunswick is provided in Figure 11, which demonstrates the trends in the use of Truck for Hire and air modes. No data were available on the use of air transportation modes. As indicated in Figure 11, the use of Truck for Hire has been increasing with a decrease in the use of rail mode.

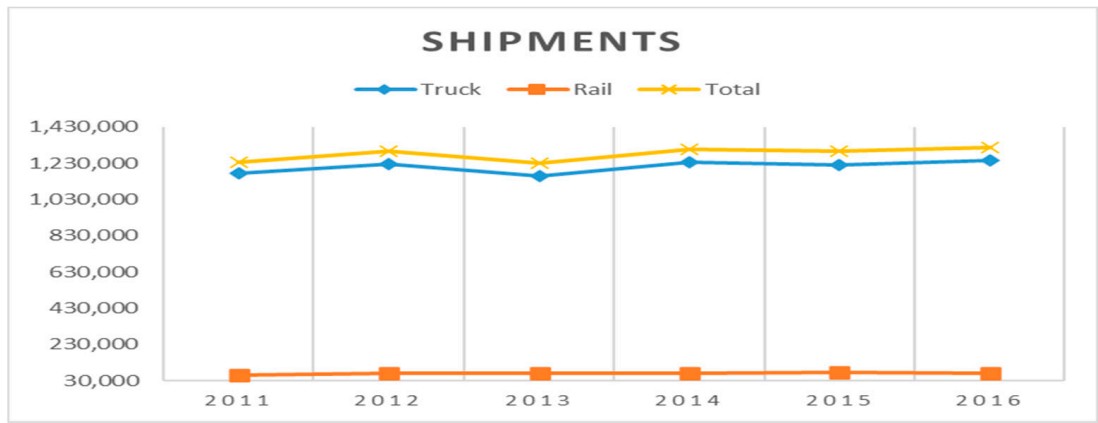

**Figure 10.** Total inbound shipments to the province of NB by transportation mode.

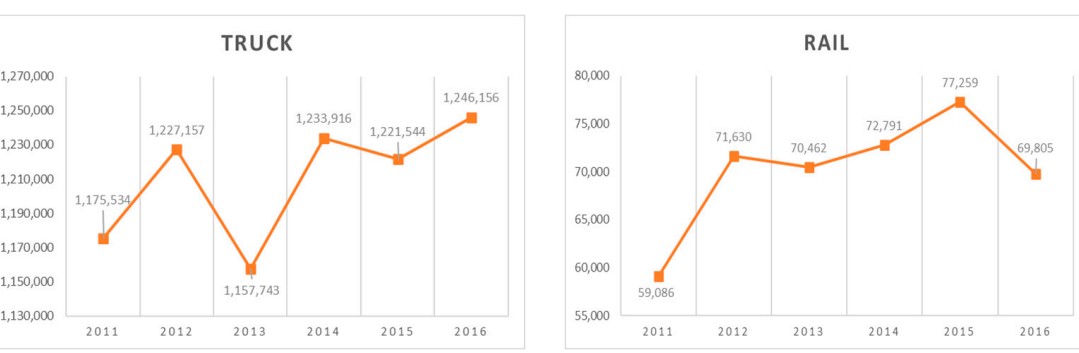

**Figure 11.** Inbound shipments to the province of New Brunswick via Truck for Hire and rails.

### 3.2.3. Newfoundland and Labrador

The total inbound shipments made by each transportation mode to the provinces of Newfoundland and Labrador (NFL) are shown in Figure 12.

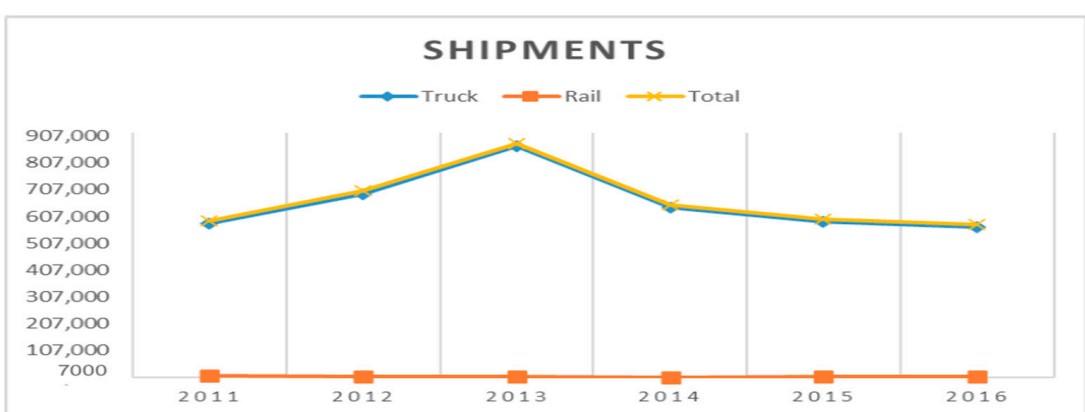

**Figure 12.** Total inbound shipments to the province of NFL by each transportation mode.

The provinces of Newfoundland and Labrador are no exception to relying heavily on the truck transportation mode for inbound shipments. In fact, 98.4% of the total inbound shipments to the provinces of Newfoundland and Labrador are made using trucks, which is higher than Nova Scotia and New Brunswick. The breakdown of the total inbound shipments to the provinces of Newfoundland and Labrador is provided in Figure 13, which demonstrates the trends in the use of truck transportation modes. Similarly to the case of the province of New Brunswick, no data were available on the use of air transportation modes in the provinces of Newfoundland and Labrador. Unlike the provinces of Nova

Scotia and New Brunswick, the total number of inbound shipments to the provinces of Newfoundland and Labrador showed a negative trend. This phenomenon is also reflected in the trend for the use of truck and rail transportation modes, i.e., both were negative. Among different possibilities, the availability of data on the use of air transportation modes could have helped justify these negative trends.

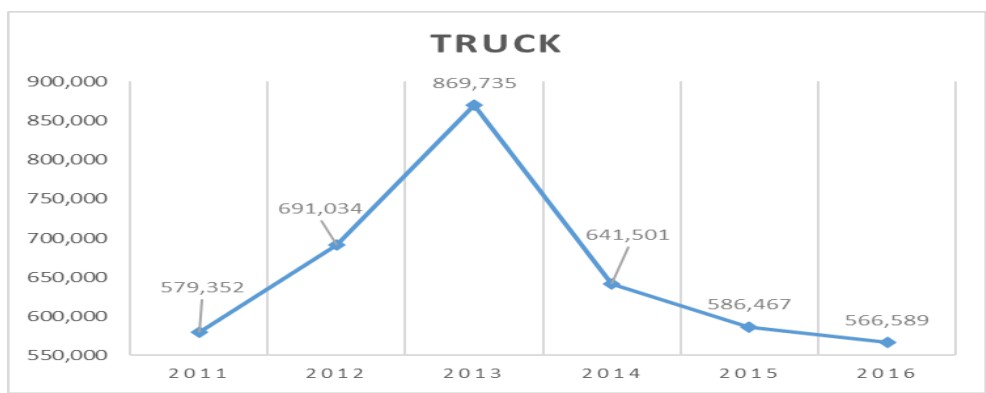

**Figure 13.** Inbound shipments to the province of NFL via Truck for Hire.

3.2.4. Prince Edward Island

The total inbound shipments made by trucks to the province of Prince Edward Island are plotted in Figure 14. No data were available on the use of air transportation mode for the province of Prince Edward Island. Additionally, rail transportation mode is not available to Prince Edward Island, so the total shipments made via trucks represent the total inbound shipments to the province of Prince Edward Island (see Figure 14).

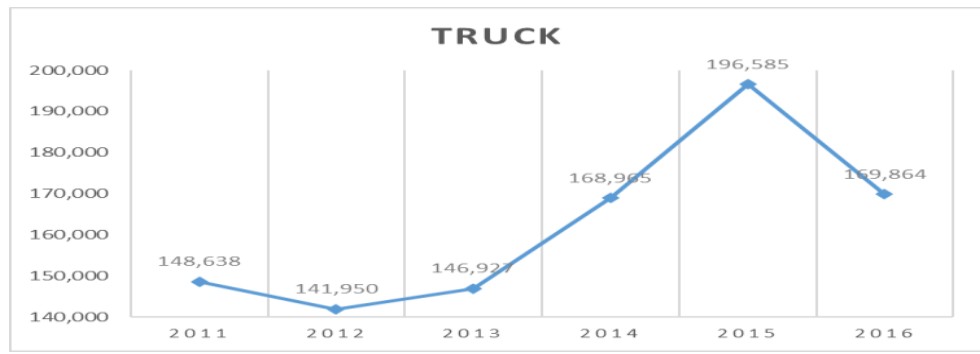

**Figure 14.** Inbound shipments to the province of Prince Edward Island via Truck for Hire.

The above analysis sheds light on the extent to which Atlantic provinces heavily rely on Truck for Hire mode. The increase in using truck service in Nova Scotia could be attributed to the closing of the Cape Breton railroad after 135 years [7].

*3.3. Relationship between Product Type and Transportation Mode*

Figure 15 depicts the transportation mode deployed to ship a variety of goods (in tons), including minerals (MNRLS), plastic and chemical products (PLCHM), forest products (FRPAP), base metals (BMETL), manufactured goods (OTHMF), automobiles and other transportation equipment (TRANS), and miscellaneous products (MISC). Trucks are used for most commodities. Additionally, trucks from Atlantic provinces are utilized to export manufactured products (OTHMF) the most. Miscellaneous items are the only goods that are exported via all three transportation modes.

Figure 15 displays the relationship between product type and mode of transport. As shown in the figure, miscellaneous items are exported the most via air, minerals are exported the most via rail, and food items are exported the most via trucks.

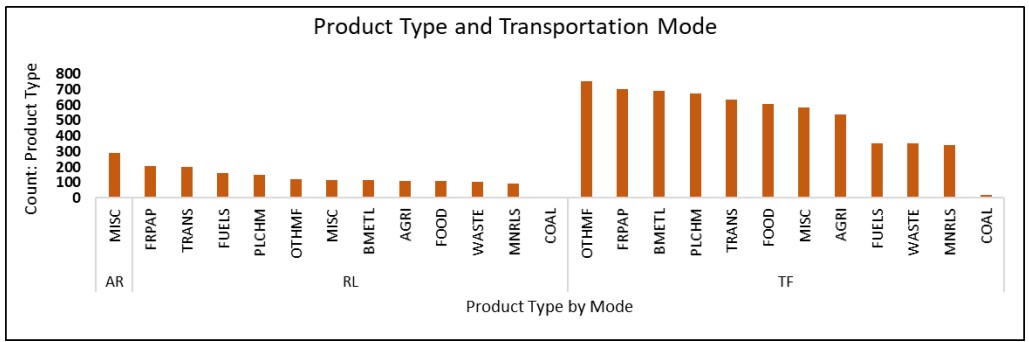

**Figure 15.** Relationship between product type and transportation mode.

### 3.4. Analysis of Freight Based on the Product Type

As shown in Figure 16, other manufactured goods, miscellaneous items, and food products are the major items imported to Atlantic provinces. The top two product categories exported from the Atlantic provinces to the rest of Canada are minerals and food products.

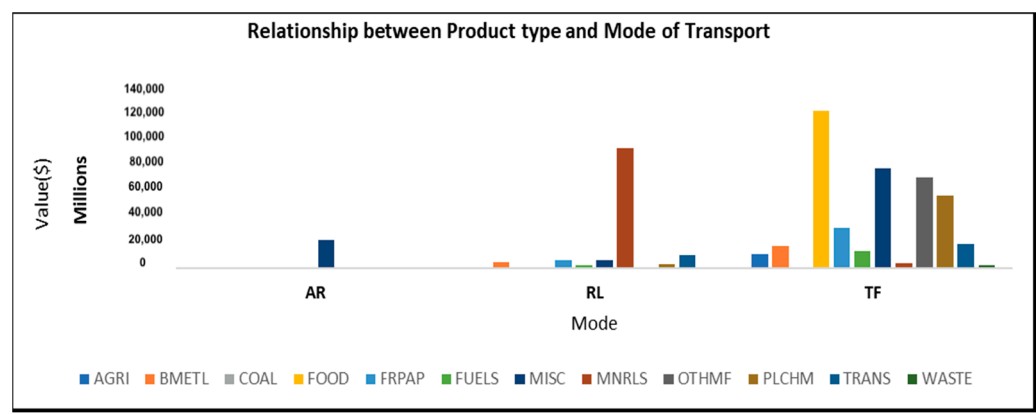

**Figure 16.** Relationship between product type and transportation mode.

Figure 17 displays the major product types exported from the Atlantic region to US and Mexico. As shown in the figure, in 2011, forest products were the major product type, but in 2014, the trend changed, and food items became the top exported items to the US and Mexico.

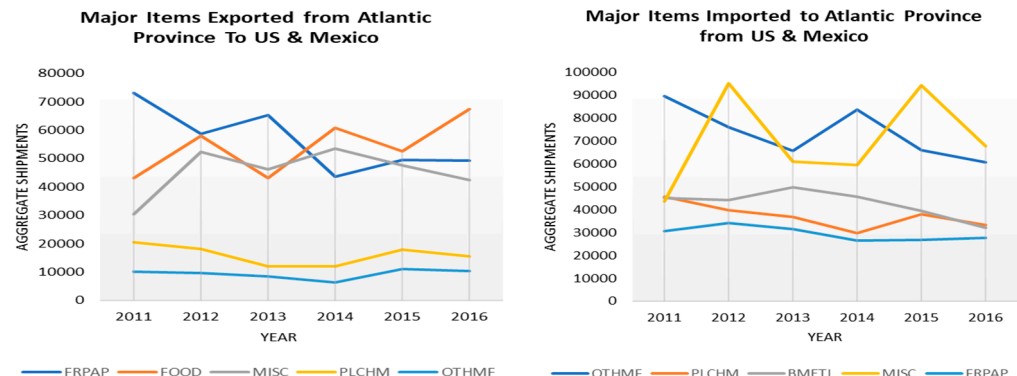

**Figure 17.** Goods exported from the Atlantic region to the rest of US and Mexico.

Minerals not only accounted for the highest volume of shipments from Atlantic provinces to the rest of Canada but also generated the highest aggregate value in shipments (see Figure 18). Similarly, miscellaneous items were the most imported products to Atlantic provinces from the rest of Canada (see Figure 18).

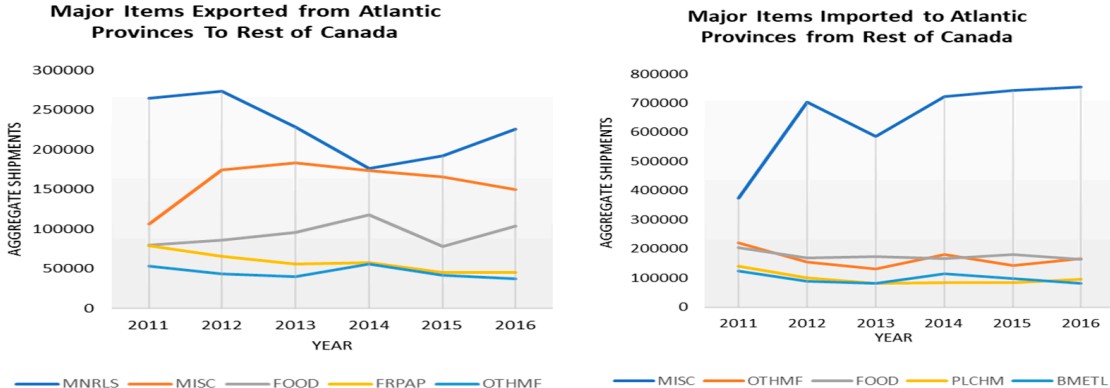

**Figure 18.** Goods exported and imported from the Atlantic region to the rest of Canada.

Figure 19 depicts the growth in the exports of fuels and automobiles from the Atlantic region to the rest of Canada. As shown in the figure, the export of fuels, automobiles, and other transportation equipment had exponential growth after 2014. This analysis excluded exports within the Atlantic provinces.

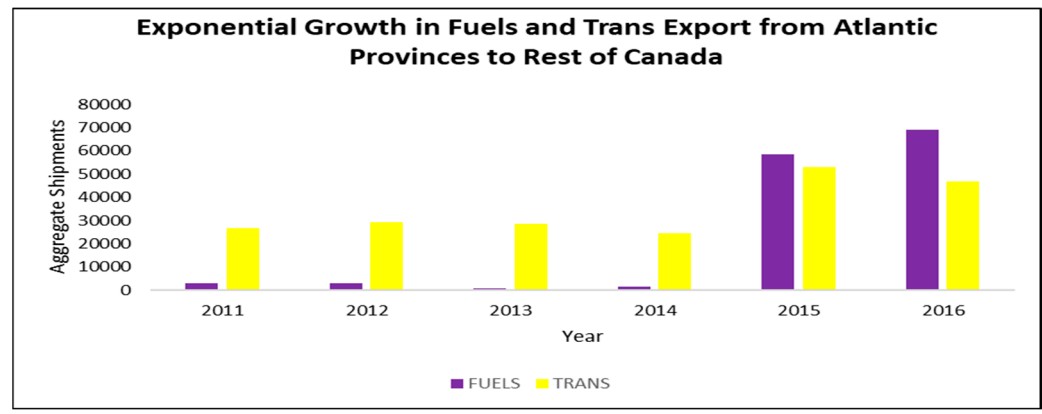

**Figure 19.** Goods exported from the Atlantic region to US and Mexico.

Figure 20 displays shipments between Atlantic provinces based on the types of goods. As shown in the figure, food products, followed by forest products and miscellaneous goods, were the main export items from Halifax, NS, to NB, NL, and PEI. It is important to note that exporting food products from Halifax and NS to other Atlantic provinces showed a decreasing trend, unlike exporting forest products and miscellaneous goods. The two major product categories imported to Halifax, NS, from other Atlantic provinces were food (with a decreasing trend) and miscellaneous goods (with an increasing trend).

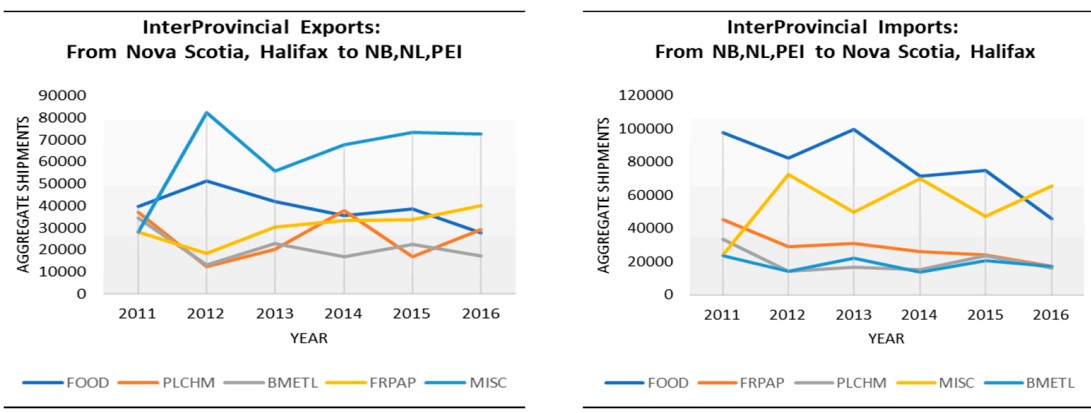

**Figure 20.** Shipments between Atlantic provinces based on good type.

Miscellaneous items accounted for the highest aggregate value in shipments (see Figure 21). This analysis excludes domestic trade within the Atlantic provinces.

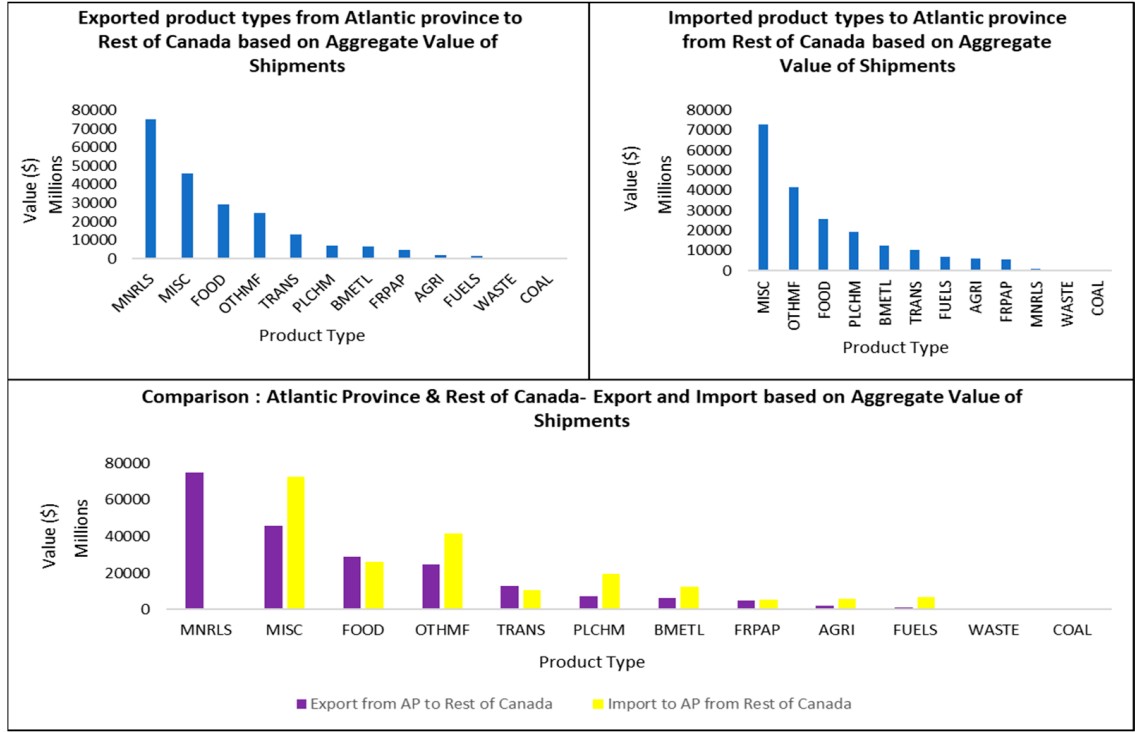

**Figure 21.** Export and import of products between Atlantic province and the rest of Canada.

Figure 22 depicts the domestic shipments between Atlantic provinces. As shown in the figure, food products are the most traded items in the Atlantic provinces.

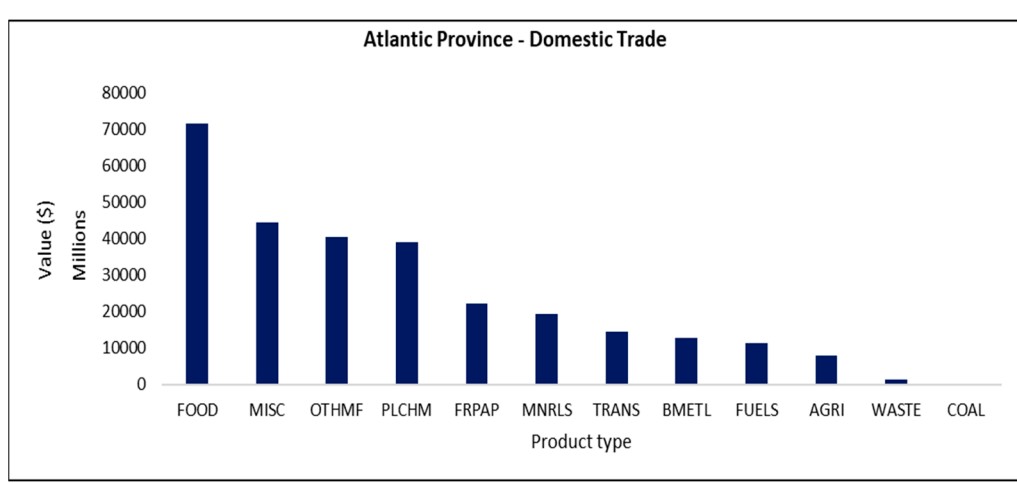

**Figure 22.** Domestic trade within Atlantic provinces (e.g., NB to NB, NB to NL, PEI to NB, etc.).

Figure 23 depicts the aggregate shipment volumes from Atlantic provinces to the rest of Canada, the US, and Mexico. As shown in the figure, the top five product categories exported from the Atlantic provinces to the rest of Canada, the US, and Mexico are miscellaneous items, food products, forest products, minerals, and other manufactured goods in that order. Accordingly, the top five export revenue-generating product categories for Atlantic provinces collectively are food products, miscellaneous items, minerals, other manufactured goods, and plastic and chemical products (see Figure 24).

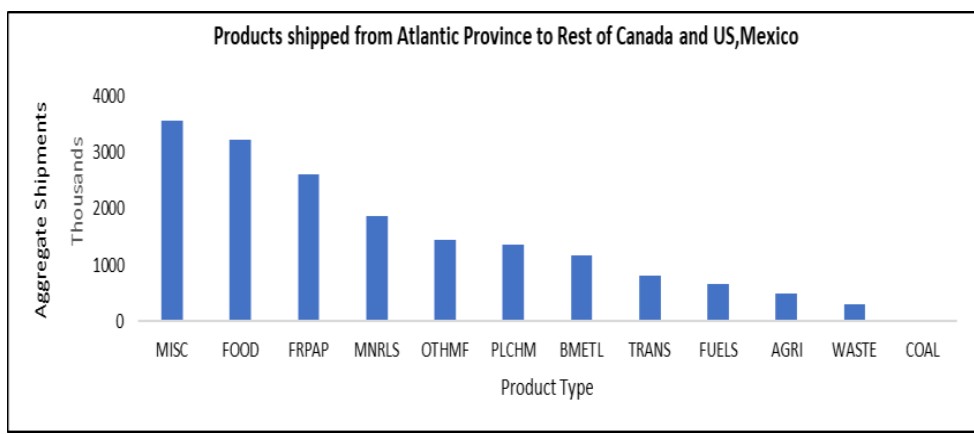

**Figure 23.** Goods shipped from Atlantic provinces to the rest of Canada and US, Mexico.

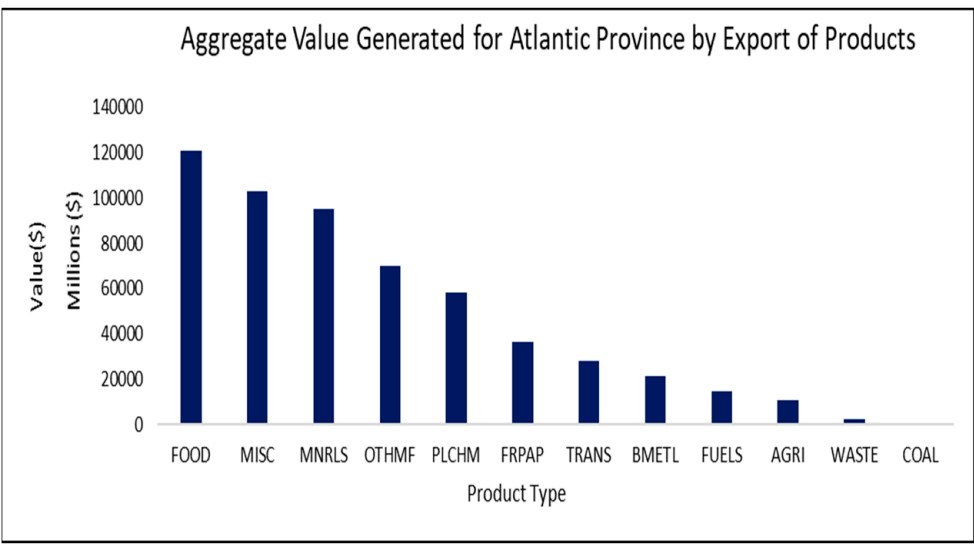

**Figure 24.** Revenue generated from exports of Atlantic provinces.

*3.5. Simple and Multiple Linear Regression Analysis*

A linear regression model was used to plot the individual relationships between the weight of commodities, distance, and the number of shipments with revenue generated.

As shown in Figure 25, revenue and weight have a positive relationship. One unit of weight increases the average revenue by 2.8%. The model predicts revenue with 50% accuracy. Thus, we can say that 50% of the variations in revenue are influenced by variations in the weights of commodities.

The linear regression model developed using distance as an input variable and revenue as the target variable showed a positive relationship between them (see Figure 26). The model was able to provide an accuracy of 50% using simple linear regression.

The linear regression model developed using revenue and number of shipments resulted in an accuracy of 42%. A unit increase in the number of shipments increased the average revenue by CAD 400 (see Figure 27).

The single-variable linear regression models did not yield high accuracy. A multiple linear regression model was developed to estimate the relationship between weight, distance, and the number of shipments with revenue generated (see Figure 28).

In contrast to the individual simple linear regression models, the R-squared value is rather high for the multiple regression model, indicating that the three independent variables together have a strong relationship with revenue with 77% accuracy.

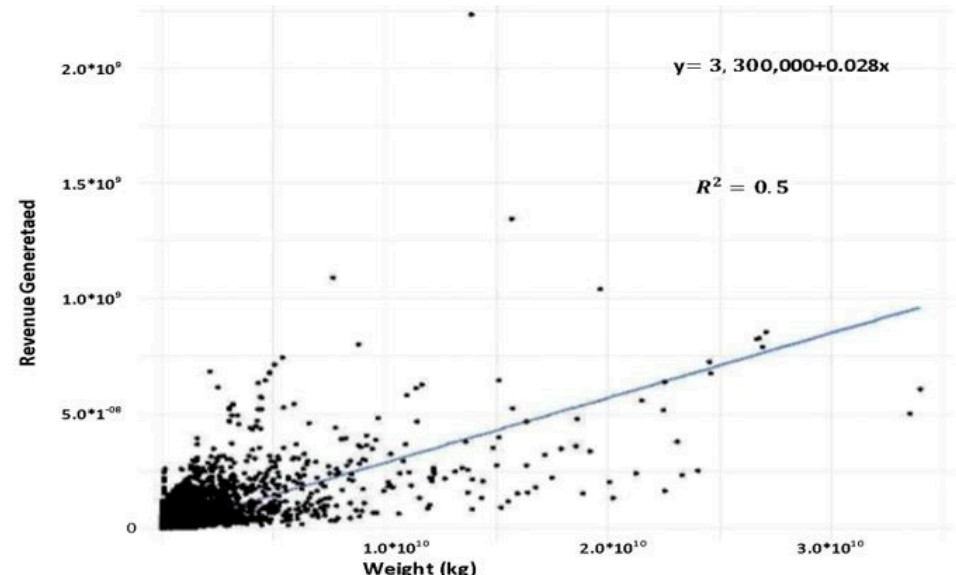

| | Estimate | Standard Error | T value | P value |
|---|---|---|---|---|
| Intercept | $3.262*10^6$ | $9.338*10^4$ | 34.93 | $2*10^{-16}$ |
| Shipments | $2.818.10^{-2}$ | $1.173*10^{-4}$ | 240.17 | $2*10^{-16}$ |

**Figure 25.** Linear regression model (weight vs. revenue).

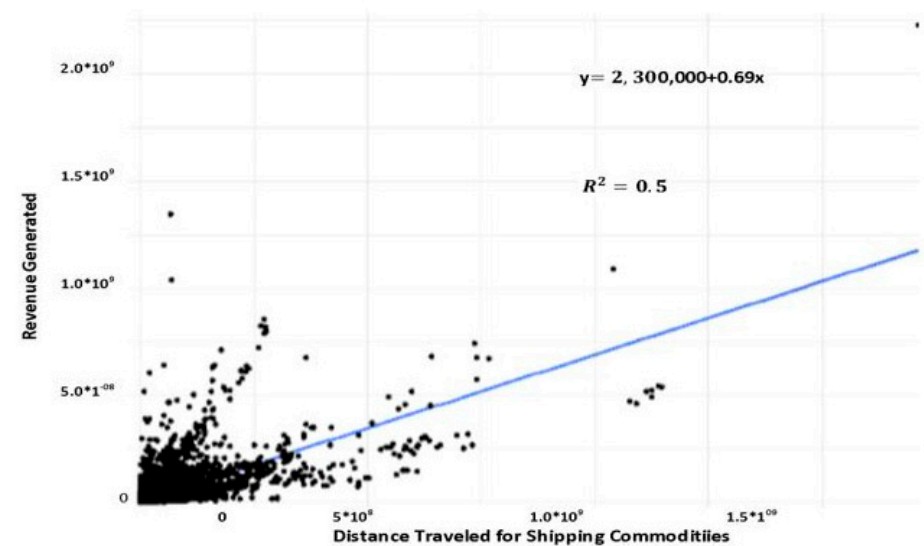

| | Estimate | Standard Error | T value | P value |
|---|---|---|---|---|
| Intercept | $2.261*10^6$ | $9.439*10^4$ | 23.96 | $2*10^{-16}$ |
| Shipments | $6.866.10^{-1}$ | $2.875*10^{-3}$ | 238.84 | $2*10^{-16}$ |

**Figure 26.** Linear regression model (distance vs. revenue).

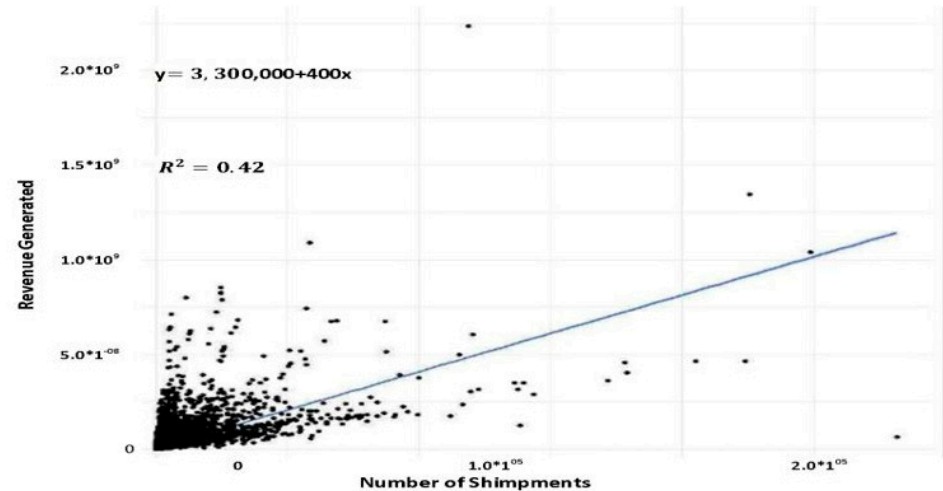

|  | Estimate | Standard Error | T value | P value |
|---|---|---|---|---|
| Intercept | $3.278*10^6$ | $1.01*10^4$ | 32.36 | $2*10^{-16}$ |
| Shipments | $4.048*10^2$ | 2.002 | 202.17 | $2*10^{-16}$ |

**Figure 27.** Linear regression model (number of shipments vs. revenue).

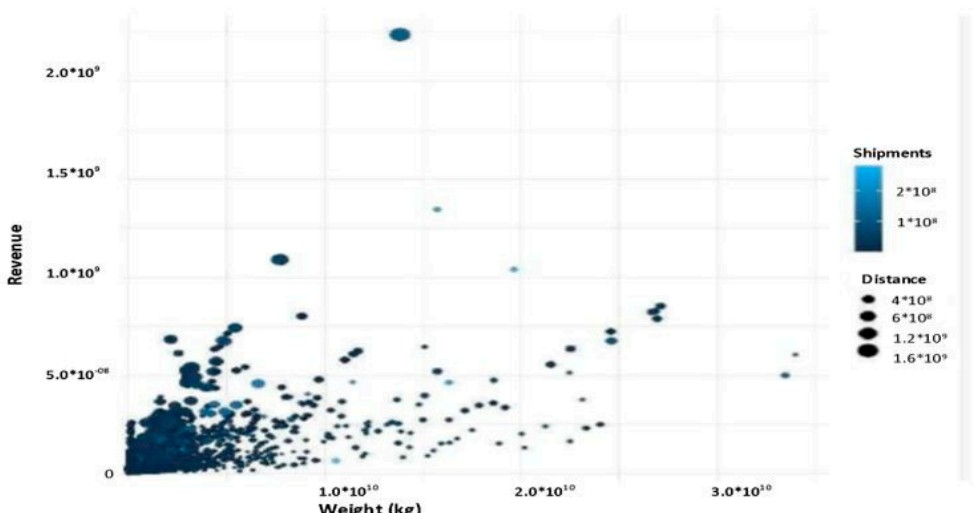

|  | Estimate | Standard Error | T value | P value |
|---|---|---|---|---|
| Intercept | $5.798*10^5$ | $6.411*10^4$ | 9.043 | $2*10^{-16}$ |
| Weight | $1.943*10^{-2}$ | $1.021*10^{-4}$ | 190.346 | $2*10^{-16}$ |
| Shipments | $6.355*10$ | 1.732 | 36.693 | $2*10^{-16}$ |
| distance | $4.946*10^{-1}$ | $2.197*10^{-3}$ | 225.062 | $2*10^{-16}$ |

**Figure 28.** Multiple linear regression model.

## 4. Conclusions

This paper presented an exploratory analysis of the Canadian Freight Analysis Framework. This is important in the context of trade and investments in the regional transportation infrastructure, as pointed out in the Atlantic Canada Transportation Strategy report [2]. Specifically, this study explored the freight transportation data for Atlantic Canada provinces with a focus on identifying the trends in transportation activities in terms of transportation modes, product types, revenue generated, and shipment values.



The findings shed light on the extent to which Atlantic provinces rely on different transportation modes and the movement of certain product types.

In this regard, we found that the top five product categories exported from Atlantic provinces to the rest of Canada, the US, and Mexico are miscellaneous items, food products, forest products, minerals, and other manufactured goods in that order. Furthermore, the top five export revenue-generating product categories for Atlantic provinces collectively are food products, miscellaneous items, minerals, other manufactured goods, and plastic and chemical products.

Minerals not only accounted for the highest volume of shipments from the Atlantic provinces to the rest of Canada but also generated the highest aggregate value in shipments. Similarly, miscellaneous items are the most imported products to Atlantic provinces from the rest of Canada and account for the highest aggregate value in shipments.

The analysis of the relationship between different product types and modes of transportation revealed that miscellaneous items are exported the most via air, minerals are exported the most via rail, and food items are exported the most via trucks. Truck for Hire is the most widely used transportation mode by the Atlantic provinces over the years. More specifically, Truck for Hire accounted for 80% of shipments, 19% of goods are shipped by rail, and 1% is shipped by air.

A simple linear regression model was used to plot the individual relationships between the weight of commodities, distance, and the number of shipments with revenue generated. In contrast to individual simple linear regression models, the R-squared value is rather high (i.e., R = 77%) for the multiple regression model, indicating that the three independent variables (weight, distance, and number of shipments) together are positively and rather strongly correlated with revenue generation.

One of the limitations of this study was the lack of data on the deployment of air and sea transportation modes in some areas. However, the findings of the study on the relationship between different product types and the mode of transportation support historical knowledge [1].

There are several directions for future research in this area. First, machine learning models could be trained and compared to predict the mode choice. Second, more explanatory variables could be investigated. Third, the findings of this study could be transferred to and compared with those of other provinces. Fourth, the introduction of sea mode choice will presumably change the mode choice behavior significantly. Finally, the mode choice behavior has changed throughout the pandemic, which raises the potential for further analysis.

**Author Contributions:** Conceptualization, J.Y. and S.A.; methodology, J.Y. and S.A.; validation, J.Y.; formal analysis, J.Y., A.M. and S.S.S.; writing—original draft preparation, J.Y. and S.A.; writing—review and editing, J.Y. and S.A.; visualization, A.M., A.G., A.Y. and S.S.S.; supervision, J.Y.; project administration, J.Y.; funding acquisition, J.Y. and S.A. All authors have read and agreed to the published version of the manuscript.

**Funding:** This research was funded by an internal RISE grant from Cape Breton University.

**Data Availability Statement:** Data supporting reported results can be found at https://www150.statcan.gc.ca/n1/pub/50-503-x/50-503-x2018001-eng.htm, accessed on 10 June 2022.

**Conflicts of Interest:** The authors declare no conflict of interest.

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
