# Peer review of "Multiple Linear Regression Analysis of Canada’s Freight Transportation Framework"

_logistics, 2023_

Round 1

Reviewer 1 Report

Notes:

1. The abstract should maintain the following structure: Background, methods and methodology, results.

2. The introduction is too small in scope. Please note that the chapter/subchapter should not be less than 1 page. In the introduction, the problem must be highlighted, as well as the justified basis for the preparation of the article, the stated purpose, etc. Therefore, it is recommended to combine the 2nd chapter with the 1st chapter and refine the above mentioned aspects.

3. Specify the title of 3.1

4. The sources from which they were taken or based on which they were compiled must be indicated under the figures and tables.

5. Abbreviations mentioned for the first time must have full names.

6. The results section is rich in graphical information, but lacks the insight and critical approach of the article's author(s).

7. Conclusions must be more specific. The direction of further research must also be indicated. Avoid references to literature sources in the conclusions.

8. As a scientific article, too few literature sources have been examined. The existing analysis does not justify the need for analysis of the problem and the conducted research and their results.

Author Response

Please see the attached file where we address the comments.

Best regards.

Reviewer 2 Report

The paper is well organised, following the traditional schemes, but...

There are 19 pages of description different statistical datasets, which are too short (just 5 years) and ended in 2016 (too old), it should last at least till 2019, afterwards we had COVID...

On the other hand there is no finding, no real messages. The 3 pages of discussion inherits just simple linear regressiond with rather low R values. There is no real connection. 

The only interesting point, the last calculation got just 2 sentences, without mathematical formula. What is the final calculus? Did you controll the final result?

Please improve the discussion part strongly! It would be nice to have a better view on the final estimation, e.g.: do the regression with the 80% of the dataset and check the goodness of the formula with the remaining 20%.

Author Response

(The authors gave the same response as above.)

Round 2

Reviewer 1 Report

Thanks for the corrections. There are a few observations:

1. Advice - cancel the numbering of subsections 1.1 and 1.2 in the introductory part. This note applies to subsections 2.1 and 2.2.

2. If possible, paraphrase the sentences in the conclusions where there are references to the 8th and 10th sources and delete those sources from the conclusions.

3. From my point of view, although there are few articles about the Canadian freight transportation framework, there are articles about the freight transportation framework in a general sense. Therefore, take a look at what scientists from other countries are studying on such issues, for example, and add to the list of literature sources.

Reviewer 2 Report

Thank you for the improvements. 
